# SPADA: Secure, Performant, and Distributed LLM Inference

kexin Chu , Jianchang Su , and Wei Zhang[*]

School of Computing, University of Connecticut
{kexin.chu, jianchang.su, wei.13.zhang}@uconn.edu

*Abstract*—**Large Language Models (LLMs) have achieved remarkable success across a range of applications, from code generation to conversational AI. As LLMs grow in size and capability, distributed inference across multiple computing nodes becomes necessary to meet resource demands and performance goals. However, this shift introduces critical security challenges, particularly in the handling of sensitive user inputs and intermediate model states like key-value (KV) caches. In this paper, we present SPADA—a Secure, Performant, and Distributed Architecture for LLM inference that addresses the core challenges of secure execution, inter-node trust, and efficient communication in distributed environments. SPADA integrates trusted execution environments (TEEs), a decentralized trust establishment protocol, and a lightweight, encrypted communication pipeline. It also introduces a secure and efficient mechanism for transmitting distributed KV cache data. Our design ensures that distributed inference pipelines maintain strong privacy guarantees without sacrificing throughput or latency, offering a practical foundation for secure LLM deployment at scale.**

## I. INTRODUCTION

Large Language Models (LLMs) have emerged as transformative technologies powering applications like open-domain question answering, code generation, document summarization, and multi-turn conversational AI [3], [7], [20]. As these models grow in size and complexity—spanning hundreds of billions of parameters—the computational and memory overhead for inference has increased significantly [8], [9], [12], [23], [36]. To support real-time applications, substantial effort has gone into performance optimization. Techniques like mixed-precision inference, model quantization, speculative decoding, and key-value (KV) cache reuse are crucial to scaling LLM inference. The adoption of high-performance hardware accelerators like GPUs, TPUs, and custom inference chips has improved throughput and latency. However, as LLM-based applications demand longer contexts and greater interactivity, these optimizations are nearing their limits, especially given the finite memory and bandwidth of a single compute node. Distributed LLM inference—partitioning the model and its state across multiple nodes—has become necessary to scale workloads efficiently [1], [15], [17], [38], [41].

While distributed inference offers performance benefits, it introduces critical security and privacy challenges [6], [19], [29], [33]. Decentralizing inference creates new attack surfaces: untrusted nodes may tamper with intermediate results, infer sensitive input prompts via timing or access patterns, or impersonate other nodes [33], [35], [45]. KV caches, storing intermediate attention keys and values for multi-turn interactions [16], [31], [39], [44], often contain sensitive information [14], [18], [21], [25], [43]. Transmitting these caches without protection poses significant risks. Research has primarily focused on performance and scalability [2], [5], [10], [11], [13], [22], [26], [30], [32], [34], [42], leaving security gaps. This issue is amplified in multi-tenant or cloud-hosted environments, where infrastructure is shared among untrusted parties, and sensitive data may be exposed or leaked [24].

Secure distributed inference requires addressing several design challenges. Participating nodes must verify trustworthiness before exchanging sensitive model states or user data. Without guarantees, compromised nodes could disrupt execution or exfiltrate information. Strong intra-node isolation is essential to preserve model integrity and confidentiality, particularly with co-located workloads or privileged software. Inter-node communication must be encrypted and authenticated to prevent eavesdropping, with minimal overhead [37]. Meeting these goals without degrading performance, especially for latency-sensitive applications, remains a complex problem.

To address these challenges, we present **SPADA** — a **S**ecure, **P**erformant, and **D**istributed **A**rchitecture for LLM inference. SPADA ensures end-to-end security for distributed inference without compromising throughput or latency. It leverages Trusted Execution Environments (TEEs), such as Intel SGX and AMD SEV, to isolate the inference runtime, model parameters, and KV cache within each node, ensuring privileged software (e.g., hypervisors or OS kernels) cannot access or tamper with sensitive state. SPADA introduces the **Decentralized Trust Establishment Protocol (DTEP)** — a lightweight protocol enabling mutually authenticated trust across distributed TEEs without a centralized authority. This design supports scalable deployments where nodes can join or leave the system dynamically.

SPADA also incorporates a high-performance secure communication layer optimized for cross-node KV cache exchange. This layer ensures confidentiality and integrity with authenticated encryption while minimizing serialization and transmission overhead. To enhance performance, SPADA supports parallelized cache fetches and pipelined token execution, ensuring security mechanisms do not become a bottleneck.

Despite these strengths, SPADA introduces new engineering

---

* Corresponding Author.

challenges. TEEs are memory-constrained and may introduce syscall or paging overheads that affect inference speed. Designing a decentralized trust protocol robust to node churn, failure, or compromise requires careful engineering and secure key management. Synchronizing KV caches across enclaves requires an efficient strategy to maintain consistency and avoid latency spikes or deadlocks.

**Our contributions are as follows:**

- We identify key privacy and security challenges in distributed LLM inference, including threats from KV cache sharing, inter-node communication, and untrusted infrastructure.
- We propose SPADA, a new architecture that integrates TEE-based isolation, decentralized trust establishment, and secure inter-node communication to ensure both security and performance.
- We design and implement DTEP, a novel trust protocol that supports dynamic, scalable trust relationships across secure inference nodes.
- We develop a secure KV cache transport pipeline that introduces minimal overhead while providing strong privacy guarantees.

## II. BACKGROUND AND MOTIVATION

### A. LLM Inference

Large Language Models (LLMs) based on the Transformer architecture use dot-product attention to compute Query (Q), Key (K), and Value (V) embeddings that capture token relationships. Inference is divided into two phases: prefill and decoding. In the prefill phase, the input prompt is tokenized and processed in parallel to compute and store K/V embeddings as the KV cache. In the decoding phase, tokens are generated autoregressively; each new token's Q attends to cached K/Vs, avoiding recomputation. Without KV caching, inference would have quadratic time complexity, making long prompts and multi-turn interactions impractical. KV reuse is thus essential for low-latency, high-throughput inference in real-time applications.

### B. TEE

Trusted Execution Environments (TEEs) are secure areas within modern processors that provide isolated execution environments to protect the confidentiality and integrity of data and code. TEEs operate by creating hardware-enforced boundaries that prevent unauthorized access from other software on the same system, including the operating system, hypervisor, or even physical attackers with root privileges. Technologies such as Intel Software Guard Extensions (SGX) and AMD Secure Encrypted Virtualization (SEV) enables sensitive computations to run securely within these enclaves, shielding them from a wide range of threats. TEEs are particularly valuable in cloud and edge computing settings, where resources are shared across multiple tenants and trust assumptions are limited. By offering hardware-based isolation, TEEs enable secure processing of sensitive data without requiring trust in the underlying system software or infrastructure.

### C. Motivation and Challenges

TABLE I: Personal Information Counts in C4 and Pile [4].

| Personal Information Type | C4 | Pile |
|---|---|---|
| User Name | 1,444,683,066 | 3,273,163,949 |
| Phone Number | 19,592,273 | 23,191,595 |
| Email Number | 9,056,833 | 13,336,793 |
| US Bank Number | 7,139,838 | 69,763,678 |
| Credit Card Number | 61,405 | 741,815 |
| US SSN | 2,352,339 | 12,541,022 |
| IP Address | 1,890,090 | 14,975,663 |
| Total | 1,484,780,621 | 3,407,722,116 |

As shown in Table I, the analysis of the C4 and Pile corpora reveals an abundance of personally identifiable tokens [27]——over 1.44 billion user names, nearly 20 million phone numbers, 9 million email addresses, 7 million U.S. bank numbers, and millions more credit-card numbers, SSNs and IP addresses. Such pervasive distribution of sensitive data imposes stringent security requirements on any LLM serving platform.

Designing a secure, high-performance, and distributed LLM inference system requires overcoming several tightly intertwined and non-trivial challenges, each of which plays a critical role in ensuring the system's overall effectiveness. The successful execution of LLM inference in a distributed environment depends on the interplay between security, performance, and scalability, all of which need to be addressed in a seamless manner. The following challenges highlight the key areas that must be tackled to build a reliable and efficient distributed LLM inference system.

- **Inter-node Trust Establishment:** A fundamental challenges in a distributed LLM inference system is establishing mutual trust among participating nodes before exchanging sensitive information. In such setting, nodes may share user-specific data—including KV cache entries and intermediate embeddings—beyond merely model parameters. In the absence of a verifiable trust mechanism, compromised or or malicious nodes can impersonate peers, manipulate inference states, or exfiltrate private data. These risks are exacerbated in cross-domain deployments, where nodes span heterogeneous administrative boundaries and are exposed to threats such as man-in-the-middle or spoofing attacks. The lack of a centralized trust authority further complicates secure coordination. Therefore, establishing decentralized and verifiable trust relationship between nodes are essential. This trust mechanism are foundational to ensure authenticity, confidentially, and integrity across the inference pipeline.
- **Intra-node Isolation and Protection:** Even within trusted nodes, security risks persist due to untrusted co-resident workloads or adversarial software. The LLM inference process exposes privacy-sensitive artifacts—such as user prompts, attention maps, and generated tokens—that are vulnerable to memory scraping, privilege escalation, or side-channel leakage. In these context,

enforcing strong intra-node isolation is critical. Trusted Execution Environments (TEEs) offer a promising solution to these issues by providing an isolated environment for sensitive computation. However, while TEEs offer strong security guarantees, they come with certain limitations, particularly in terms of memory capacity, I/O bandwidth, and programming constraints. For resource-intensive applications like LLM inference, ensuring that the TEEs can operate efficiently while maintaining strong security guarantees is a major challenge. The balance between securing the model execution and managing the resource constraints of TEEs is essential to maintain high-performance inference while safeguarding sensitive data.

- **Secure and Low-Overhead Communication:** In a distributed LLM inference system, the communication between nodes must be secure to ensure the protection of sensitive data such as user inputs, intermediate results, and cached context embeddings. As these communications often involve highly sensitive user information, strong encryption and authentication mechanisms are necessary to prevent unauthorized access, eavesdropping, or data manipulation. However, implementing robust security protocols should not come at the expense of system performance. Low-latency communication is essential in real-time LLM inference systems to ensure fast response times and high throughput. The challenge, therefore, lies in finding a balance between securing data transmission and minimizing the performance overhead associated with encryption and authentication. This requires the design of lightweight communication protocols that can maintain the security of data exchanges without introducing significant delays. The efficiency of these protocols becomes even more critical as the scale of the distributed system grows, as each additional node introduces more potential points of communication and, consequently, greater risk of performance bottlenecks.

- **Secure and Efficient Transmission of Distributed KV Cache:** A critical component of distributed LLM inference is the efficient management and sharing of KV caches across multiple nodes. The KV cache plays a pivotal role in optimizing inference performance by storing intermediate states such as attention-key (K) and attention-value (V) embeddings. These caches are essential for maintaining context across multiple inference steps, especially in long-running or multi-turn interactions. However, since KV caches may contain sensitive user data or context from previous interactions, their secure transmission between nodes is paramount. Without proper protection, there is a risk of data leakage during the transfer of KV cache entries. Ensuring that the KV cache data is encrypted and transmitted securely across distributed nodes while maintaining performance is a delicate balance. The process of encrypting and securely transmitting KV cache entries must not introduce significant delays or computational overhead, as this would undermine the overall performance of the system. Effi-

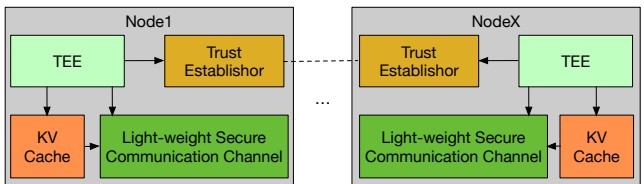

Fig. 1: Overview of the SPADA architecture

cient serialization, lightweight encryption, and optimized transmission protocols are necessary to ensure that KV cache data is transmitted swiftly and securely, enabling continuous context sharing across nodes while upholding strong privacy guarantees. This challenge is particularly important because the KV cache is a key performance optimization for LLM inference, and any inefficiency in its secure transmission could significantly degrade the system's overall performance.

### D. Threat Model

The threat model for distributed LLM inference focuses on protecting sensitive data and model states from a range of potential security risks, including unauthorized access, eavesdropping, and data tampering. Key threats include malicious or compromised nodes attempting to intercept or alter intermediate results [40], leakage of sensitive user inputs via unprotected KV caches [28], and attacks on the communication channels between nodes [25], [31], [43]. To mitigate these risks, the system employs encryption and authentication for secure data transmission, TEEs to isolate model parameters and data, and decentralized trust protocols to ensure secure, mutually authenticated node interactions. These measures aim to maintain data confidentiality, integrity, and privacy while ensuring robust performance in distributed settings.

### III. SYSTEM DESIGN

To meet the stringent requirements of secure, performant, and distributed LLM inference, we propose **SPADA**—a **S**ecure, **P**erformant, and **D**istributed **A**rchitecture that tightly integrates TEEs, decentralized trust mechanisms, and optimized, enclave-aware communication protocols. SPADA is designed to mitigate the unique threats and performance constraints introduced by distributing large-scale LLM inference across potentially untrusted infrastructure. This section presents the core components of SPADA, each addressing a major challenge identified in secure and scalable LLM inference: trust establishment among nodes, intra-node protection, secure inter-node communication, and privacy-preserving KV cache transfer, shown in figure 1.

### A. Decentralized Inter-node Trust Establishment

Distributed inference pipelines rely on the frequent exchange of sensitive intermediate states—such as KV cache entries, embeddings, and token predictions—across physically or administratively disparate nodes. In such environments, establishing mutual trust is a prerequisite for secure collaboration. Traditional centralized trust models based on PKI

or certificate authorities are ill-suited for federated or elastic deployments. SPADA introduces a **Decentralized Trust Establishment Protocol (DTEP)** that leverages TEE-backed remote attestation to verify node authenticity without relying on central infrastructure.

Each node in SPADA is provisioned with a hardware-backed attestation capability (e.g., Intel SGX's EPID or ECDSA attestation, AMD SEV's attestation report), which cryptographically proves the integrity of its runtime and workload to remote peers. During protocol bootstrapping, participating nodes exchange signed attestation evidence and verify each other's enclave identity, configuration, and measurement hash. This process establishes a shared root of trust across nodes operating under different domains or cloud providers. Using enclave-internal Diffie-Hellman key exchange, SPADA securely derives symmetric session keys that are never exposed outside the enclave. This secure key material underpins all subsequent data transfers and prevents unauthorized nodes from participating in the distributed inference graph.

### B. Intra-node Isolation and TEE-based Protection

Even with secure inter-node communication, inference workloads remain vulnerable to a wide spectrum of intra-node threats—including privilege escalation, memory snooping, and malicious co-tenants. To mitigate these risks, SPADA leverages TEEs to enforce **strong isolation guarantees within each node**, safeguarding sensitive inference artifacts such as user inputs, attention scores, and generation history from other co-located processes.

SPADA encapsulates the entire model execution pipeline—including token embedding, Transformer blocks computations, and KV cache management—within a hardware-isolated enclave. This design ensures that adversaries with root or hypervisor-level access cannot inspect or tamper with internal inference states. However, TEEs inherently impose strict constraints on memory footprint, execution models, and I/O access due to their limited secure resources.

To address these challenges, SPADA incorporates a suite of memory- and compute-efficient techniques: quantized operator pipelines reduce model size, activation checkpointing mitigates memory pressure, and enclave-local cache eviction policies manage the constrained secure memory effectively. Additionally, SPADA employs batched enclave calls and asynchronous execution pipelines to reduce context switch overhead and maximize parallelism. Together, these optimizations enable SPADA to deliver low-latency, privacy-preserving inference while operating within the limitations of secure enclaves.

### C. Secure and Low-Overhead Inter-node Communication

LLM inference is particularly sensitive to communication latency due to its sequential generation pattern and the size of intermediate state, especially in decoder-only Transformers. SPADA addresses this with a **secure and latency-optimized communication layer** that integrates cryptographic protections directly into the enclave execution flow.

SPADA terminates TLS sessions inside the enclave using enclave-compatible libraries such as WolfSSL or Rustls-TEE, ensuring that both session keys and plaintext payloads are confined within trusted boundaries. Unlike traditional models where TLS termination occurs in untrusted user space or OS layers, SPADA's approach eliminates a wide class of man-in-the-middle and key-exfiltration attacks. To reduce communication overhead, SPADA employs a binary protocol with fixed-length headers and aligned memory buffers, enabling zero-copy data transfer from enclave memory to the network stack. This design avoids unnecessary serialization or memory duplication that would otherwise incur overhead within the constrained TEE memory region. Furthermore, to defend against traffic analysis, SPADA pads messages and aggregates them into burst-mode packets, obfuscating timing and payload size correlations.

### D. Secure and Efficient Distributed KV Cache Transmission

At the core of Transformer-based LLM inference lies the KV cache, which stores past attention keys and values and enables the model to process each token in constant time during decoding. In a distributed execution model, where attention heads or layers are partitioned across nodes, transmitting KV cache blocks becomes essential. However, most of these caches come from user' prompt, which contains highly sensitive informations, including user privacy information, system prompts, etc., which must be protected during transmission.

SPADA introduces a **secure and bandwidth-efficient cache propagation mechanism** that encrypts all KV cache fragments using per-session keys derived during mutual attestation. Each transmitted segment is appended with a unique nonce, sequence number, and MAC, enabling strict detection of tampering and replay attacks. To reduce communication volume, SPADA incorporates delta encoding of KV cache blocks, transmitting only the subset of keys and values that have changed between decoding steps. This is particularly effective for sparse updates common in multi-turn conversations and autoregressive decoding. Additionally, SPADA applies token-level attention predictions to preemptively prefetch relevant cache blocks, overlapping communication with computation and reducing perceived latency. Together, these optimizations allow SPADA to securely propagate cache state with minimal impact on throughput, even as context lengths scale into thousands of tokens.

## IV. CONCLUSION

The rising computational demands of LLM inference have made distributed execution a necessity. Yet, this distribution expands the attack surface and creates an urgent need for security primitives that protect sensitive user data and model states without compromising performance. In this work, we introduced SPADA, a secure, performant, distributed inference architecture that integrates trusted execution environments, decentralized trust protocols, and low-overhead secure communication. Through mechanisms—including mutual attestation, enclave-based model execution, encrypted cache pipelines,

and delta-aware KV cache transmission—SPADA achieves strong privacy guarantees while maintaining responsiveness for interactive applications. While implementation remains ongoing, SPADA lays the groundwork for a scalable approach to secure LLM inference, paving the way for trustworthy deployment of large-scale AI models in federated, heterogeneous environments.

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
