

# SPADA: Secure, Performant, and Distributed LLM Inference

**Kexin Chu, Jianchang Su, Wei Zhang**

School of Computing, University of Connecticut

**Contact Information:**
233 Glenbrook Rd, Unit 4100,
Storrs, CT 06269-4100, USA

Phone: +1 959-995-3930
Email: kexin.chu@uconn.edu

**Abstract**

As Large Language Models (LLMs) scale in size and application scope, distributed inference across multiple computing nodes becomes essential to meet latency and memory demands. However, this introduces new risks to privacy and trust, especially due to the exposure of sensitive inputs and intermediate states like KV caches. We present SPADA: a Secure, Performant, and Distributed Architecture that integrates Trusted Execution Environments (TEEs), a Decentralized Trust Establishment Protocol (DTEP), and secure KV cache transport to address these concerns. SPADA ensures end-to-end privacy protection without sacrificing throughput or latency, making it a practical foundation for secure LLM deployment in untrusted or multi-tenant settings.

## Introduction

LLMs power applications from code generation to chat systems, but their inference cost has surged due to growing model size and context length. Techniques like KV caching and hardware acceleration are widely used, yet often insufficient under single-node memory constraints. Distributed inference improves performance but increases vulnerability to tampering, data leakage, and node impersonation. Current efforts mainly target scalability; SPADA instead focuses on the security-performance trade-off by introducing a system that safeguards privacy during distributed LLM inference.

## Motivation

Modern LLMs are increasingly deployed in sensitive, user-facing contexts—chatbots, virtual assistants, healthcare, education, and finance—where privacy and data security are paramount. While distributed inference is a practical necessity due to resource constraints, it dramatically increases the attack surface for leaking sensitive data. SPADA is motivated by the urgent need to secure LLM inference pipelines in such settings.

### Real-world Risk: Personal Information Leakage

Table I in the paper presents a striking analysis of two widely used training datasets: C4 and The Pile. It demonstrates the staggering prevalence of Personally Identifiable Information (PII) in raw LLM training data. When these models are deployed for inference, such PII can resurface in generated outputs or linger in KV caches, posing a severe privacy threat.

| Personal Information Type | C4 | Pile |
|---|---|---|
| User Name | 1,444,683,066 | 3,273,163,949 |
| Phone Number | 19,592,273 | 23,191,595 |
| Email Number | 9,056,833 | 13,336,793 |
| US Bank Number | 7,139,838 | 69,763,678 |
| Credit Card Number | 61,405 | 741,815 |
| US SSN | 2,352,339 | 12,541,022 |
| IP Address | 1,890,090 | 14,975,663 |
| Total | 1,484,780,621 | 3,407,722,116 |

**Table 1:** Personal Information Counts in C4 and Pile.

## Challenges & Solutions

### Inter-node Trust Establishment

⚠ **Problem:** In distributed inference, nodes must exchange sensitive data (e.g., KV caches, embeddings) across untrusted domains.

🔒 **Risk:** Without verifiable trust, malicious nodes may impersonate others, manipulate model states, or exfiltrate private user inputs.

🔒 **SPADA Solution:**
- Introduces the **Decentralized Trust Establishment Protocol (DTEP)**
- Uses TEE-backed attestation to verify node identity and configuration
- Establishes shared session keys via in-enclave Diffie-Hellman exchange
- Eliminates need for centralized trust authorities

### Intra-node Isolation and Protection

⚠ **Problem:** Even within a trusted node, adversarial software or co-tenants may target sensitive inference data.

🔒 **Risk:** Prompts, attention maps, and generation outputs can be exposed via memory scraping or side-channel attacks.

🔒 **SPADA Solution:**
- Runs entire inference pipeline inside **Trusted Execution Environments (TEEs)**
- Shields sensitive state from OS, hypervisors, and co-resident processes
- Mitigates TEE resource limits using:
  - Quantized operators
  - Activation checkpointing
  - Batched enclave calls

### Secure and Low-Overhead Communication

⚠ **Problem:** Inter-node communication involves transmission of sensitive states and requires encryption without introducing latency bottlenecks.

🔒 **Risk:** Traditional TLS outside TEEs may expose plaintext and increase latency.

🔒 **SPADA Solution:**
- Performs TLS termination **inside TEEs** using enclave-compatible libraries
- Employs a **binary protocol** with fixed headers and zero-copy buffers
- Obfuscates traffic patterns using **padding and burst-mode aggregation**

### Secure and Efficient Transmission of Distributed KV Cache

⚠ **Problem:** KV caches contain private token representations and must be exchanged across nodes during inference.

🔒 **Risk:** Unprotected cache transfer can leak sensitive prompts or be tampered with in-flight.

🔒 **SPADA Solution:**
- Encrypts KV cache fragments using per-session keys derived from attestation
- Adds nonce, sequence number, and MAC for integrity and replay protection
- Applies **delta encoding** to reduce transmission volume
- Uses **token-level prefetching** to overlap communication and computation

## Overview of SPADA

**SPADA** delivers secure, high-throughput distributed LLM inference by tightly integrating:

- **TEE-based Node Isolation**: Each node runs inference inside a *Trusted Execution Environment*, ensuring sensitive states (e.g., KV cache, embeddings) are invisible to OS and co-tenants.
- **Decentralized Trust (DTEP)**: Nodes perform *remote attestation* to mutually verify enclave identity and derive per-session encryption keys—no centralized trust anchor needed.
- **Secure Communication**: All inter-node traffic flows through a *lightweight encrypted channel*, with *in-enclave TLS termination* and traffic padding to resist metadata leakage.
- **Efficient KV Cache Transfer**: SPADA sends only the updated portions of KV cache via *delta encoding*, authenticated with nonces and MACs, and prefetches cache blocks to overlap compute and I/O.

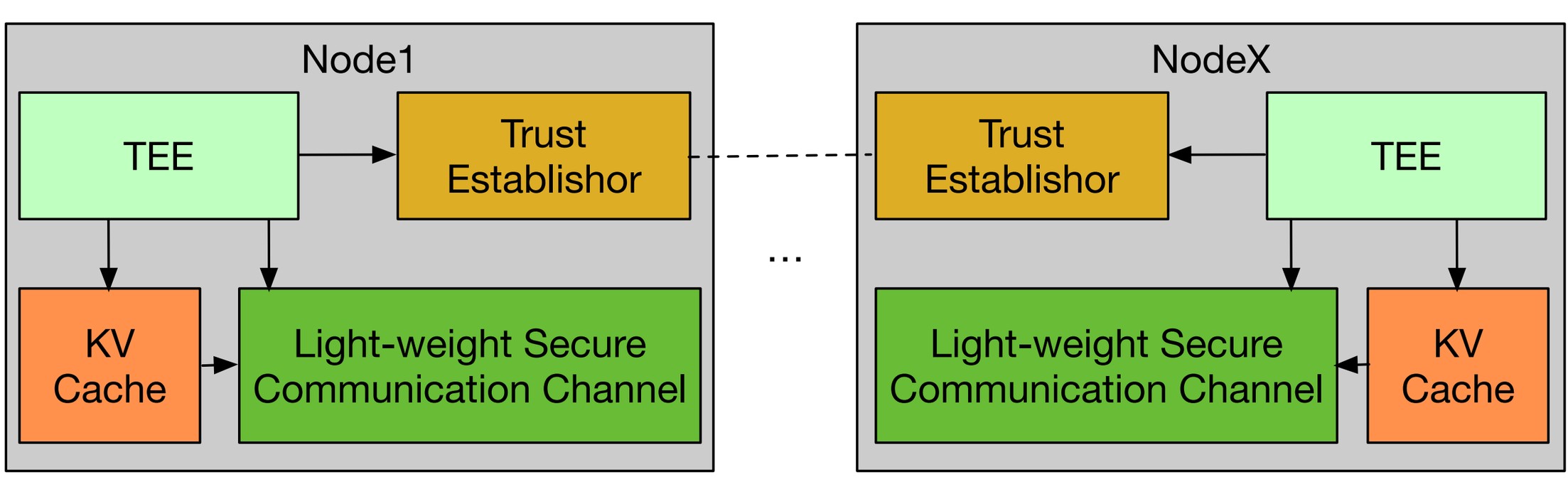

**Figure 1:** Overview of the SPADA architecture

## Conclusions

- SPADA secures the entire distributed LLM inference lifecycle—from node authentication to KV cache transfer.
- By combining TEEs, DTEP, and optimized communication, it defends against inference leakage without slowing down inference.
- We demonstrate that privacy and performance are not mutually exclusive in large-scale LLM deployments.

## Future Work

- Support heterogeneous TEEs (e.g., Intel SGX + AMD SEV + ARM CCA)
- Integrate with memory-efficient LLMs (e.g., LoRA, MoE, quantized models)
- Extend SPADA to support confidential training across distributed TEEs