# OpenReview forum: "SPADA: Secure, Performant, and Distributed LLM Inference"
_iscaconf.org/ISCA/2025/Workshop/MLArchSys — MLArchSys 2025 Poster_

### Official Review · Reviewer_RbYv · 2025-05-18
**No Design Evaluated**

**Confidence:** 4
**Rating:** 1

**Detailed Feedback And Questions For Authors:**

To be more compelling, this paper should teach something to the reader that the reader doesn't already know. TEEs and encryption for distributed systems are a known technique, and latency is a well-known evaluation metric. Several critical questions that this paper leaves unanswered:

1. What latency is sufficient for LLM inference? How is this latency measured (median, 99th percentile, something else)? I.e., how will you know once you've succeeded at this project?
2. For a security work, what is your threat model?
3. Why are current systems inadequate? SPADA builds on TEEs and a rich body of distributed systems and cryptography. Given this, what about existing systems is inadequate for LLM inference?
4. What did you build, and how did you build it? For software systems, are there any unique software engineering techniques that you used? What tradeoffs did you encounter, or design decisions?
5. How well does SPADA perform relative to the prior work? How does it perform relative to the sufficient latency metrics?

Additionally, the paper is unclear from the introduction as to what is actually implemented. For example, the end of the introduction includes "we design and implement DTEP" and "we develop a secure KV cache transport pipeline", but the conclusion states that "implementation remains ongoing"

**Top Reasons To Accept The Paper:**

None

**Top Reasons To Reject The Paper:**

This paper proposes SPADA, but the authors do not appear to have actually implemented or tested SPADA or a prototype. The paper uses a lot of buzzwords (TEE, attestation, etc.), but the authors do not bring these concepts together into an integrated system design proposal. Furthermore, the authors propose requirements, like latency constraints, without discussing specific quantitative limits.

---

### Official Review · Reviewer_Q9yp · 2025-05-18
**A Secure, Performant, and Distributed Architecture for LLM called SPADA**

**Confidence:** 3
**Rating:** 3

**Detailed Feedback And Questions For Authors:**

This paper had a promising start, and the design seems to make sense from a thousand foot view. But without results, it really is difficult to vouch for it or against it. I'd encourage the authors to continue working in this area, as the design objectives seem reasonable.

**Top Reasons To Accept The Paper:**

Important problem, good motivation, and good discussion of design choices and efficiency approaches.

**Top Reasons To Reject The Paper:**

No results. The design seems ok, but without results it really is difficult to validate any of the design decisions or claims made.

---

### Official Review · Reviewer_XUSm · 2025-05-19
**SPADA: Integrated Security Framework for Distributed LLM Inference**

**Confidence:** 3
**Rating:** 3

**Detailed Feedback And Questions For Authors:**

Thanks for submitting the paper.

It was interesting to read the need for secure distributed LLM systems. It would have been beneficial if some more discussion on how the constraints differ in LLM systems compared to traditional systems.
Some minor suggestions:
1. Table 1 seems unnecessary and unrelated motivation for distributed LLM systems.
2. The indentation of sections in Section IIC takes up a lot of space - it would be helpful to remove the indentation and add more details in the paper.

**Top Reasons To Accept The Paper:**

1.  The paper proposes SPADA which claims to be one of the first comprehensive architectures specifically designed to address the security challenges of distributed LLM inference.
2. The authors describe the security challenges of current distributed LLM inference systems in detail.

**Top Reasons To Reject The Paper:**

1. Lack of Implementation Details: The authors describe a conceptual architecture without providing concrete implementation details. This seems more like a position paper - a more comprehensive analysis of the available solutions in each part and the pros and cons of each could have provided a tutorial value if a complete implementation was not in scope.
2. The novelty of the paper seems to be from building an end-to-end conceptual system - it was hard to identify the exact novelty in the design of SPADA. It would be beneficial to compare why traditional approaches of building secure systems would fail in an LLM distributed systems.
3. Some discussion on partitioning mechanism in accelerators such as MiGs in GPUs was expected in the intra-node execution.

---

### Official Review · Reviewer_aEHq · 2025-05-19
**This paper identified the challenges in privacy and security of today's LLM inference serving systems and presents a design of SPADA.**

**Confidence:** 4
**Rating:** 3

**Detailed Feedback And Questions For Authors:**

Thank you for submitting your work to MLArchSys'25. The paper brings up key privacy and security challenges, discusses their implications, and challenges in building a solution for it.

While the paper does a great job of making a case for the need for such a framework, it doesn't quite show the initial results. I appreciate the design of the proposed LLM inference architecture, SPADA. But, I was hoping ot see some initial results of SPADA. Without an initial signal on how well it works, what metrics of success are considered, what tetsbed, what models, workload is considered, it is really hard to find if the proposed approach has the potential to work. I strongly recommend building an initial prototype.

**Top Reasons To Accept The Paper:**

- The paper is well written ad focuses on a practical problem.

**Top Reasons To Reject The Paper:**

- The paper in its current form isn't ready for publication, as the paper only presents the design without any preliminary evaluation of the peoposed design.